# Seasonal Surface Fluctuation of a Slow-Moving Landslide Detected by Multitemporal Interferometry (MTI) on the Huafan University Campus, Northern Taiwan

Chiao-Yin Lu [1,2], Yu-Chang Chan [3,*], Jyr-Ching Hu [1], Chia-Han Tseng [4], Che-Hsin Liu [2] and Chih-Hsin Chang [2]

1   Department of Geosciences, National Taiwan University, Taipei 10617, Taiwan;
    d05224001@ntu.edu.tw (C.-Y.L.); jchu@ntu.edu.tw (J.-C.H.)
2   National Science and Technology Center for Disaster Reduction, Taipei 23143, Taiwan;
    a120160@ncdr.nat.gov.tw (C.-H.L.); chang.c.h@ncdr.nat.gov.tw (C.-H.C.)
3   Institute of Earth Sciences, Academia Sinica, Taipei 11529, Taiwan
4   Department of Geology, Chinese Culture University, Taipei 11114, Taiwan; zjh17@ulive.pccu.edu.tw
*   Correspondence: yuchang@earth.sinica.edu.tw; Tel.: +886-2-27839910 (ext. 1411)

**Abstract:** A slow-moving landslide on the Huafan University campus, which is located on a dip slope in northern Taiwan, has been observed since 1990. However, reliable monitoring data are difficult to acquire after 2018 due to the lack of continuous maintenance of the field measurement equipment. In this study, the multitemporal interferometry (MTI) technique is applied with Sentinel-1 SAR images to monitor the slow-moving landslide from 2014–2019. The slow-moving areas detected by persistent scatterer (PS) pixels are consistent with the range of previous studies, which are based on in situ monitoring data and field surveys. According to the time series of the PS pixels, a long period gravity-induced deformation of the slow-moving landslide can be clearly observed. Moreover, a short period seasonal surface fluctuation of the slow-moving landslide, which has seldom been discussed before, can also be detected in this study. The seasonal surface fluctuation is in-phase with precipitation, which is inferred to be related to the geological and hydrological conditions of the study area. The MTI technique can compensate for the lack of surface displacement data, in this case, the Huafan University campus, and provide information for evaluating and monitoring slow-moving landslides for possible landslide early warning in the future.

**Keywords:** slow-moving landslide; seasonal surface fluctuation; gravity-induced deformation; multitemporal interferometry (MTI); Huafan University campus



## 1. Introduction

Landslide hazards happen more frequently in areas where surface cracks or slow-moving activity already exist than in other areas [1–3]. A slow-moving landslide generally does not cause direct injury to human life, but it can cause direct damage to buildings or facilities on the surface, which may bring about additional social and economic costs [4–6]. However, slow-moving landslides can evolve into rapid and destructive landslides, which have caused extensive damage to buildings and threatened human lives globally in recent decades [7–10]. According to the Emergency Event Database (EM-DAT), approximately seventeen thousand people were killed by 349 catastrophic landslides around the world from 2000–2017. In Taiwan, based on the database of the National Science and Technology Center for Disaster Reduction (NCDR), over 11,000 slope failures and landslides occurred, which caused 1033 deaths from 2000–2017. Surface displacement data can be used to characterize the boundary and the activity of a slow-moving landslide. Therefore, efficiently detecting and monitoring slow-moving landslides can provide better clues to identify potential landslide sites over a large area. The monitoring results are crucial for landslide hazard risk management and early warning.

The location and scale of slow-moving landslides could be preliminarily determined by features of topography based on remote sensing images and with other supporting information such as geologic maps, slope aspects, in situ monitoring data and field surveys [11–14]. However, difficulties remain for the detailed characterization of slow-moving phenomena by in situ observations over greater spatial and temporal scales due to their poor spatial resolution and deployment limitations. In addition, field measurements are rather labor intensive and costly. From the viewpoint of landslide hazard assessments, long period monitoring to provide possible early warning of unstable slopes by exploiting affordable remote sensing data is a very significant issue. The first application of synthetic aperture radar differential interferometry (DInSAR) to detect an unstable slope showed the capability of InSAR-related techniques to detect and monitor slow-moving landslides [15]. The development of the multitemporal interferometry (MTI) technique overcomes the main limitations, including coherence loss in vegetated areas and atmospheric effects, which limit the performance of DInSAR in landslide investigations [16–18]. Several studies have focused on applying the MTI technique for detecting slow-moving landslide areas, monitoring slope deformation, and analyzing the time series of the displacements of a slow-moving landslide over a long period of time [3,16,18–22]. The hydrology-driven acceleration of a slow-moving landslide in the wet season by using low temporal revisited SAR images, such as those from the ALOS and ENVIST satellites, has been described in previous studies [23–25]. The capability of the MTI technique to detect and monitor slow-moving landslides can be enhanced by increasing the revisit frequency by approximately a few days, improving the resolution of space-borne sensors and expanding the spatial coverage by up to hundreds or thousands of square kilometers. Thus, cost-effective and high-precision monitoring data of the surface displacements over a large area over a long period of time can be obtained [21,26].

In Taiwan, where hills and mountains occupy approximately 70% of the area, infrastructure and private properties are often constructed in mountainous regions due to the growing population, expansion of settlements and economic requirements. The Huafan University campus, with 3700 faculty and students, is located on a dip slope in northern Taiwan. The dip slope has been investigated to be unstable with fractures on the ground, roads and structures since the campus was established in 1990 [27–29]. A monitoring system has been gradually established since 2000 to address safety concerns. However, because the maintenance of equipment and measurement of data require considerable money and manpower, reliable monitoring data were difficult to collect after 2018. This study applied the MTI technique with free Sentinel-1A/B SAR images to monitor and analyze the surface displacement of a slow-moving landslide from 2014–2019. The results of the MTI technique not only can provide information of surface displacement in different aspects, but also can compensate for the lack of in situ monitoring data in the study area. The combination and calculation of the results derived from the ascending and descending SAR images can reveal the vertical and E-W displacement velocity fields. The long period gravity-induced deformation of a slow-moving landslide can be observed using the time series of the PS pixels. Moreover, the seasonal surface fluctuation of a slow-moving landslide can be clearly detected from the time series in this study. A seasonal interaction model [30], which was proposed in our previous study, can be used to interpret this phenomenon. To test a stable monitoring point for the PSInSAR analysis in this area, a corner reflector (CR) was designed and deployed over the northern part of the campus. The purpose of this study is to present detailed information of surface displacement and compensate for the lack of in situ monitoring data for a slow-moving landslide, in this case at the Huafan University campus, by using the MTI technique with free SAR images.

## 2. Study Area

### 2.1. Geological Setting

The Huafan University campus is constructed in the upper part of the Dalun Mountain area, which is located at the northern end of the Western Foothills belt in northern

Taiwan (Figure 1). The Dalun Mountain area is a dip slope with an average dip angle of approximately 20° toward the southwest. The elevation of the campus ranges from 450 to 570 m above sea level, and the area of the campus is approximately 34 hectares. According to geological surveys [31–33], the Miocene Mushan Formation is considered to be the dominant lithology of the study area. The bedrock is composed of sandstone (SS) and thin alternating layers of sandstone and shale (SS-SH). The bedrock is overlain by a 10–20 m thick cover layer, which is composed of regolith, colluvium and fill. Two small-scale faults were determined to pass through the study area according to the 2D resistivity data, borehole data and topographic features [32,33]. The Nanshihkeng Fault is a northeastern striking reverse fault with a dip angle of 60°–70°. Another fault called the A fault is a normal fault with a left-lateral slip component, which is truncated by the Nanshihkeng Fault. There are two sets of joints in the bedrock with average attitudes of NW–SE and NNE–SSW, which are considered to confine the boundaries of the sliding block in the study area [29]. The average annual precipitation based on rain gauge records from 2003–2010 was approximately 4000 mm, which was mainly attributed to typhoons and torrential rainfall [27]. Surface cracks have been observed and have expanded gradually on the slope surface and in some buildings due to the slow-moving phenomenon of the slope since the establishment of the campus. Settlement is especially clear in areas with thicker filled material, such as the Asoka Square, the sports ground and the basketball court [27–29].

### 2.2. Monitoring System and Failure Mechanism

To understand the failure mechanism and provide risk management, monitoring systems including 32 inclinometer casings, 32 standpipes for groundwater table monitoring and 2 rainfall gauges have been installed since 2000. There are 15 tiltmeters installed on the building walls, 48 strainmeters installed on the reinforcing bars and 36 strainmeters installed on the concrete of the buildings for monitoring the tilt situation of the structures. In addition, a network of 295 ground monitoring points was established in 2001 for measuring the sliding behavior of the dip slope using conventional traverse surveying twice a year (Figure 1a).

According to the explicit displacement of the inclinometer casings recorded monthly and after heavy rainfall and earthquake events, the depth of the sliding surface of approximately 10–40 m was revealed (Figure 1b) [28]. A plausible model of landslide movement with a listric sliding surface [29] was proposed to explain the process of sliding blocks on the campus. This model assumes that the head part of the sliding surface has a concave-upward shape and then becomes parallel to the bedding plane of the bedrock on a certain weak surface. After the sliding block departs downslope on the sliding surface and creates a gap between the head and crown, unconsolidated material collapse filling the gap shows movement toward the upslope direction from the first measurement in 2001 to the present day. This represents the sliding blocks moving with a slow velocity over an extended time, while several intermittent large slips appear during periods of heavy rainfall. Multiple potential sliding blocks that exist within the campus are revealed by the long-term surveying results. Based on the velocity field of the horizontal displacement of the ground monitoring points from 2010 to 2017, the varying displacement velocities and directions indicate that two plausible sliding blocks exist within the campus. To obtain more reliable results, methods including in situ geological investigations, borehole data and inclinometer measurements were applied and present the same range of the two sliding blocks [29]. Since the stratigraphic sequences derived from the borehole data of the campus are different, the two sliding blocks are in different geological settings. One sliding block in the southeastern part of the campus is believed to be translational deformation with a slow-moving velocity of approximately 20–30 mm/yr. This sliding surface is considered to be a listric-shaped surface within the bedrock with a depth of approximately 30–40 m. In the northwest part of the campus, another sliding block with multiple listric sliding surfaces at the head slowly

moves on the interface with a velocity of approximately 12 mm/year. The sliding surface is between the loosely consolidated colluvium and the underlying bedrock [29].

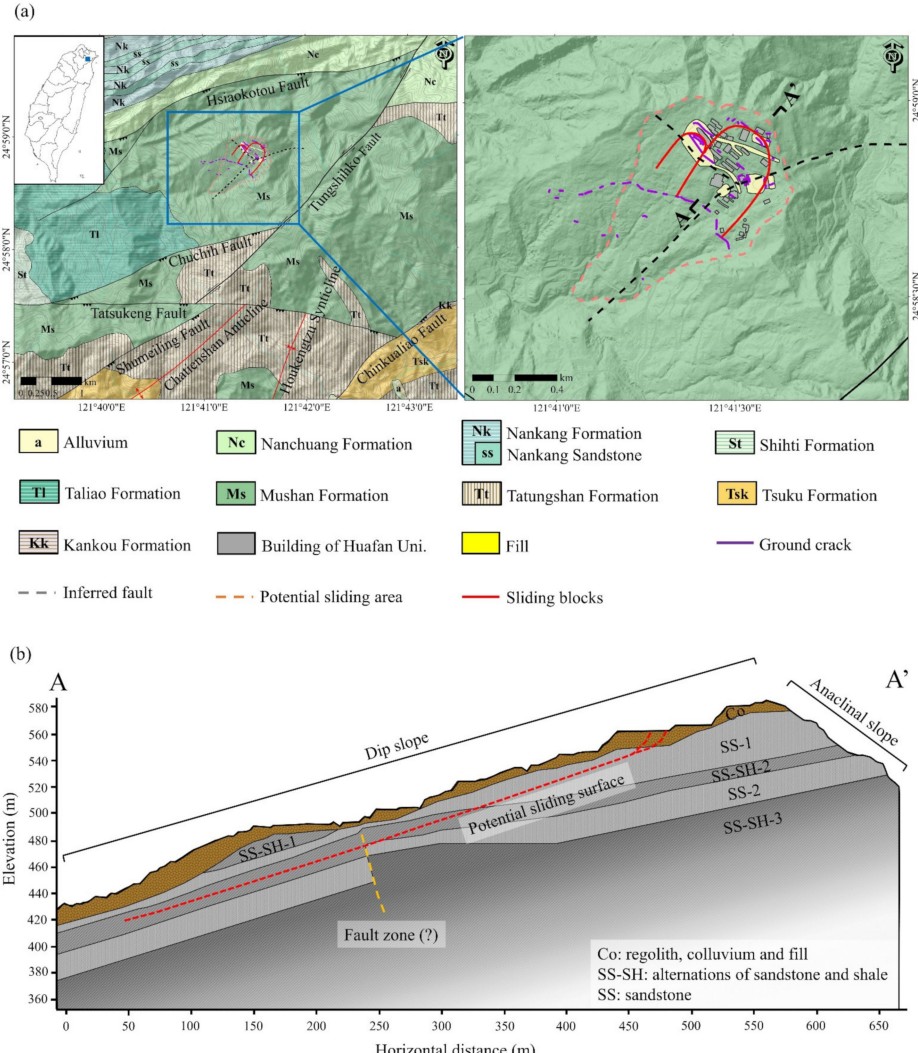

**Figure 1.** (**a**) Geological map of the Dalun Mountain area, which is located at the northern end of the Western Foothills belt in northern Taiwan. The dominant lithology of the study area is the Miocene Mushan Formation. Red lines indicate the areas of the two sliding blocks, while the orange dotted line indicates the potential landslide area determined from topographic features based on LiDAR-derived topography. The background topography is shown by shaded relief imagery processing from 20 m and 2 m digital elevation models (DEMs). (**b**) The geological profile is along line AA′, which is modified from [29]. The bedrock of this dip slope is composed of sandstone and alternations of sandstone and shale.

## 3. Methodology

### 3.1. Multitemporal Interferometry (MTI)

Multitemporal interferometry (MTI) techniques identify stable radar targets to monitor surface displacement by processing a long temporal series of SAR images, including persistent scatterer SAR interferometry (PSInSAR) [34,35], small baseline subset (SBAS) [36] and similar approaches [37]. The persistent scatter (PS) pixel is identified as the value of a pixel dominated by a stable and strong scatter that is brighter than the background scatterers. Under this condition, the underlying displacement signal can be extracted because the variance in the phase of the background scatterers is relatively small and can be ignored. The selection of the PS pixels in the conventional method [34] relies

on thresholding pixel amplitude dispersion through time, which has a high success rate at detecting bright PS pixels, such as human-made structures. The PSInSAR method developed by Hooper et al. [35] proposed a new PS selection approach based on the spatial correlation of the phase characteristics of pixels. This approach is applicable to low-amplitude natural targets with low-phase variance in all terrains, with or without buildings. These cannot be detected as PS pixels using the amplitude-based algorithms of the conventional method. Moreover, not requiring a prior model of deformation velocity is an important advantage of the PSInSAR procedure. This method relies on the spatially correlated nature of the deformation rather than requiring a known model of the temporal deformation [35]. The interferometric phase terms can be written as [35]:

$$\Phi_{int} = \Phi_{defo} + \Phi_{atmos} + \Phi_{orbit} + \Phi_{topo} + \Phi_{noise} \tag{1}$$

where $\Phi_{defo}$ is the phase change due to deformation along the line-of-sight (LOS) direction, $\Phi_{atmos}$ is the atmospheric delay, $\Phi_{orbit}$ is the orbit error, $\Phi_{topo}$ is the DEM error and $\Phi_{noise}$ is the noise due to variability of scattering. The $\Phi_{defo}$, $\Phi_{atmos}$, $\Phi_{orbit}$ and $\Phi_{topo}$ are assumed spatially correlated over distances. The PS pixel is defined when the noise term is small enough.

Thus, in this study, the PSInSAR method was applied to calculate and analyze the slow-moving phenomenon of the Huafan University campus with C-band Sentinel-1A/B images. A set of 177 Sentinel-1A/B interferometric wide swath images (single polarization VV) acquired in the ascending orbital geometry and 160 images acquired in the descending orbital geometry during the period from approximately October 2014–October 2019 were used in the PSInSAR calculation procedure. In addition, a dataset of 14 Sentinel-1A/B images acquired in the ascending orbital geometry from 29 February 2020 to 25 May 2020 was used to verify the effectiveness of the deployment of the CR. To remove the phase caused by the topography, the 30 m Shuttle Topography Mission (SRTM) digital elevation model (DEM) is applied. This study used Sentinel Application Platform (SNAP) software, which was developed by the European Space Agency (ESA), to generate interferograms of the single look complex (SLC) images from Sentinel-1. Then, we applied the widely used PSI software package StaMPS/MTI (Stanford Method for Persistent Scatterers/Multi-Temporal InSAR) [38,39] for determining the PS pixels of all interferograms. A newly developed software package, which called snap2stamps, can integrate the output of SNAP with StaMPS to perform the PSInSAR processing [40,41]. Therefore, the displacement along the radar line of sight (LOS) direction can be measured and calculated based on the phase difference of the PS pixels in the SAR images over a long period of time.

### 3.2. Calculation of the Projected LOS Velocity and 2D Displacement Velocity Field

The most representative component of a translational landslide movement is considered to be along the direction of the maximum slope. However, a PSInSAR measurement can only detect the displacement component parallel to the LOS direction. Thus, to facilitate data interpretation and compare landslide velocities with different slope aspects, we projected the LOS velocity derived from both the ascending and descending images onto the maximum slope direction as $V_{slope}$ in the study area. The formula used to calculate the value of $V_{slope}$ was modified from previous studies [42–44]:

$$V_{slope} = V_{LOS}/C \tag{2}$$

The coefficient C is calculated from the parameter slope (S) and aspect (A) in degrees derived from the DEM data, and the direction cosine of the LOS is represented as $N_{LOS}$, $E_{LOS}$ and $H_{LOS}$:

$$C = N_{LOS}(cos\ cos\ (S)\ \cdot sin\ sin\ (A-90)) - E_{LOS}(cos\ cos\ (S)\cdot cos\ cos\ (A-90)) + H_{LOS}(sin\ sin\ (S)) \tag{3}$$

$N_{LOS}$, $E_{LOS}$ and $H_{LOS}$ also represent the percentage of the real displacement vector registered along the LOS direction of the sensor and can be estimated using the following equations:

$$\begin{cases} N_{LOS} = -\cos(90 - \alpha) \cdot \cos(180 - \gamma) \\ E_{LOS} = -\cos(90 - \alpha) \cdot \cos(270 - \gamma) \\ \qquad H_{LOS} = \cos(\alpha) \end{cases} \qquad (4)$$

where $\alpha$ indicates the incident angle of the SAR sensor and $\gamma$ indicates the LOS azimuth in degrees. Table 1 shows the average value of the direction cosine of the LOS of the ascending and descending orbital geometries in this study area. For instance, the $H_{LOS}$ derived from the calculation based on descending SAR images is 0.82, which means that approximately 82% of the possible vertical displacement can be detected by the PSInSAR analysis based on the Sentinel-1 images.

**Table 1.** The percentage of possible displacement of N-S, E-W and vertical direction that can be detected by the PSInSAR analysis based on the ascending and descending Sentinel-1 SAR images.

|  | **N–S** | **E–W** | **Vertical** |
|---|---|---|---|
| *Ascending* | 15% | 67% | 73% |
| *Descending* | 12% | 56% | 82% |

The 2D displacement velocity field, including eastward and vertical motion, can be obtained based on two different satellite viewing geometries: ascending and descending. To analyze the 2D displacement field, the ascending ($vLOS_a$) and descending ($vLOS_d$) LOS velocities were interpolated using the inverse distance weighted (IDW) method with $15 \times 15$ spatial resolution in ArcGIS. Then, the equations modified from previous studies were used for calculating the 2D displacement fields, assuming that the north component is negligible [42–44]:

$$V_{eastward} = \frac{((vLOS_d/HLOS_d) - (vLOS_a/HLOS_a))}{((ELOS_d/HLOS_d) - (ELOS_a/HLOS_a))} \qquad (5)$$

$$V_{vertical} = \frac{((vLOS_d/ELOS_d) - (vLOS_a/ELOS_a))}{((HLOS_d/ELOS_d) - (HLOS_a/ELOS_a))} \qquad (6)$$

where $HLOS_a$, $HLOS_d$, $ELOS_a$ and $ELOS_d$ indicate the direction cosines of the LOS vector derived from the ascending (a) and descending (d) orbital geometries, which are calculated by the azimuth ($\gamma_a$ and $\gamma_d$) and the incidence angle ($\alpha_a$ and $\alpha_d$) in degrees:

$$\begin{cases} ELOS_a = -\cos(90 - \alpha_a) \cdot \cos(270 - \gamma_a); \ ELOS_d = -\cos(90 - \alpha_d) \cdot \cos(270 - \gamma_d) \\ \qquad HLOS_a = \cos(\alpha_a); \ HLOS_d = \cos(\alpha_d) \end{cases}$$
$$(7)$$

### 3.3. Corner Reflector Design and Deployment

According to previous studies, corner reflectors have been successfully installed in the field to increase the persistent scatterers for monitoring slope deformation using interferometric synthetic aperture radar (InSAR) techniques where natural persistent scatterers are sparse or nonexistent [45–48]. It is possible to design a network of corner reflectors based on geodetic requirements and cover the whole area of interest with arbitrary spatial density. However, there are few research results of applying corner reflectors to monitor the displacements of slow-moving landslides in Taiwan.

The most commonly used corner reflector consists of three triangular metal panels welded perpendicularly to each other to form a trihedral shape. To obtain a bright and stable response in SAR images, specific requirements are needed for the design of trihedral corner reflectors [45–48]. To reach the maximum radar cross section (RCS), the orientation of the reflectors relative to the radar should be carefully determined and measured. The

orientation of the corner reflector needs to be perpendicular to the orbit of the satellite. The radar LOS and the axis-of-symmetry of the corner reflector must be parallel, which is related to the incident angle of the SAR acquisition. As a result, the corner reflectors can effectively reflect the signal from the SAR sensor with a maximum response. In addition, the installation of corner reflectors should be avoided in shadow and layover areas. These requirements allow the corner reflector to be detected by the SAR sensor with sufficient intensity.

The minimum size of a corner reflector is a function based on the wavelength of the SAR sensor and the backscattering level of the surroundings. The RCS is a backscattering coefficient of a target, which represents the ability of the target to reflect radar signals to the radar receiver. The theoretical peak RCS value ($\sigma_{max}$) of a trihedral corner reflector is identified by the edge length of the corner reflector using the equation below [45,48,49]:

$$\sigma_{max} = 4\pi\alpha^4/3\lambda^2 \tag{8}$$

where $\lambda$ is the wavelength of the radar and $\alpha$ is the length of the leg of the right triangle. Typical backscatter levels of flat cultivated terrain with low vegetation are approximately within the range of $-10$ dB to $-14$ dB when considering a range of radar incidence angles of C-band satellites [48]. To detect a corner reflector in a C-band SAR image, the difference in the RCS between the corner reflector and the surroundings should be larger than 30 dB [45,48]. Under the conditions of using Sentinel-1 satellites with a wavelength of 5.67 cm and setting a small corner reflector with a leg length of 1 m, the maximum RCS value of the corner reflector is 30 dB, which is enough to identify the corner reflector against the vegetated surroundings.

In the present case, one trihedral corner reflector was set up in a garden of the upper part of the Huafan University campus where natural persistent scatterers are nonexistent. Since the slope faces southwest, the ascending acquisitions can provide better sensitivity to the downward displacement. Thus, the corner reflector was deployed according to the acquisition geometry of the Sentinel-1 ascending orbital geometry, in which the incident angle ($\theta$) and the direction of the LOS are approximately 43°N and 79.4°E from the study area (Figure 2). The trihedral corner reflector was manufactured by three right triangle stainless steel panels with a leg length of 1 m and was set up from 27 February 2020 to 4 June 2020.

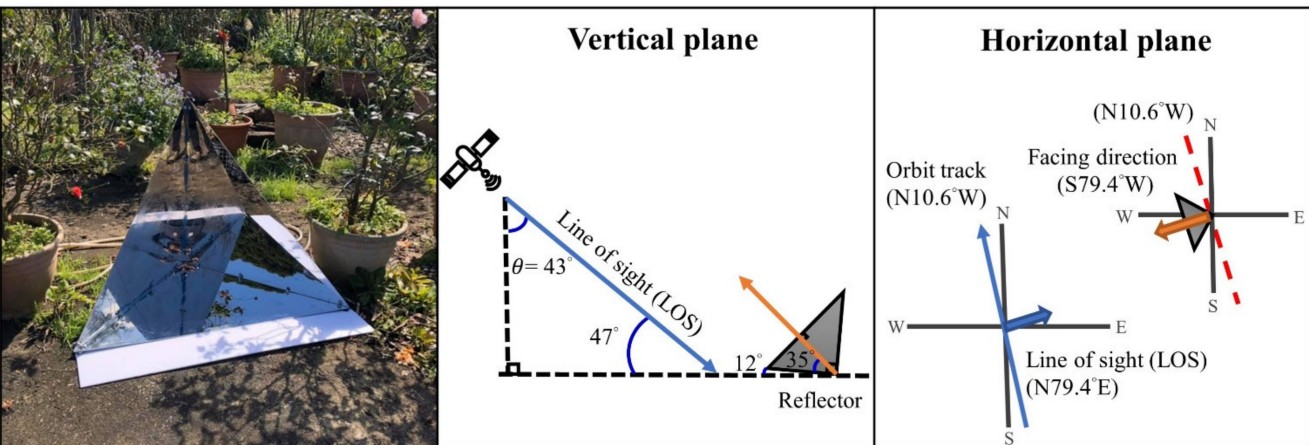

**Figure 2.** Geometry of the corner reflector designed for the ascending orbital geometry of the Sentinel-1 satellite, which was set up at the upper part of the Huafan University campus. The elevation of the corner reflector is 12 degrees and faces S79.4°W, which can effectively reflect the signal from the SAR sensor with a maximum response.

## 4. Results

For the purpose of monitoring and analyzing the slow-moving landslide at the Huafan University campus, this study applied 177 ascending SAR images from 22 October 2014 to 26 October 2019 and 160 descending SAR images from 02 November 2014 to 28 October 2019, which were all derived from the Sentinel-1A/B satellites. The dates of the master images are 28 May 2018 and 05 July 2018, which were selected according to the correlation of the temporal interval, perpendicular spatial baseline, Doppler centroid frequency baseline and thermal noise. The results of the PSInSAR method are described in detail as follows:

### 4.1. Surface Displacement Monitoring of the Slow-Moving Landslide

The slow-moving phenomena of the unstable slopes could be detected clearly using the PS pixels derived from the ascending and descending images (Figure 3). Red indicates that the displacement is away from the satellite, while blue indicates that the displacement is toward to the satellite. The results from the ground monitoring points installed after 2001 [29] around Ming-Yue Building, shown in the purple square (Figure 3), are set as the reference points during the PSInSAR analysis. These two figures both illustrate significant displacement along the LOS direction. The slow-moving area detected by the PS pixels in this study coincides with the two sliding block areas, which are defined by the previous study [27,29]. The clear slow-moving phenomenon mainly occurs at the area near points B and E in the larger sliding block, while slower movement occurs near the boundary. The analysis of the images from both the ascending and descending orbital geometries revealed a maximum displacement velocity of approximately 6.6 mm/yr along the LOS. A conservative velocity threshold that defines the state of activity of a slow-moving landslide is set by Colesabti and Wasowski [18] as ±2 mm/yr. Thus, the smallest displacement velocities of 2 mm/yr are observed largely outside the sliding blocks denoted as green circles and are considered to be the stable area in Figure 3.

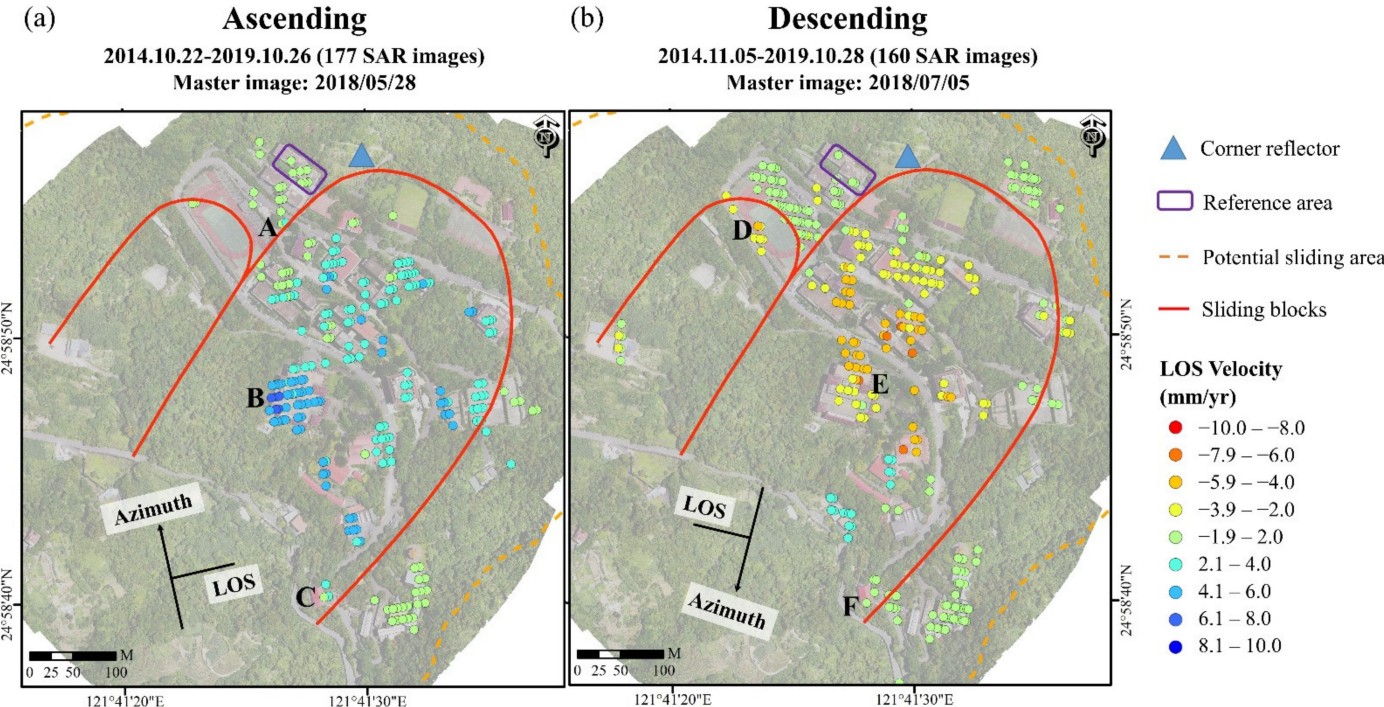

**Figure 3.** PSInSAR results of (**a**) ascending images and (**b**) descending images. The maximum displacement velocity along the LOS direction was approximately 6.6 mm/yr. A significant slow-moving phenomenon of the Huafan University campus was detected. These figures illustrate the surface displacement along the LOS direction. The red color indicates that the displacement was away from the satellite, while the blue color denotes that the displacement was toward to the satellite. Green circles indicate the relative stable area. Red lines indicate the areas of the two sliding blocks, while the orange dotted

line indicates the potential slide area determined by topographic features based on LiDAR-derived topography. The purple square represents the area of the ground monitoring points used, which were calculated as reference points for PSInSAR processing. The letters A-F indicate the positions of the selected PS pixels for plotting the time series. The background is an aerial photo taken on 04 June 2020 by NCDR.

### 4.2. The Time Series of the Selected PS Pixels

In addition, the PS pixels at different positions of the unstable slope were chosen to plot the time series from 2014 to 2019 (Figure 4). The mean value of the PS pixels in a radius of 50 m of the selected PS pixels was calculated to evaluate the surface displacement over time. The time series of PS pixels A, B and C were derived from the results based on the ascending images, and D, E and F were derived from the results based on the descending images. After February 2018, Sentinel-1 SAR images can be acquired every 6 days in Taiwan, which enhances the continuity of the time series. According to the time series of PS pixels B and E, the clear slow-moving trend in the middle of the potential landslide area can be derived. The maximum accumulated deformation of the slope during the monitoring period could reach approximately 20–30 mm along the LOS direction. The green dashed line presents the long period surface displacement trend of the slow-moving phenomenon. The positive slope of the green dashed line indicates that the displacement is toward to the satellite in the LOS direction, and the negative slope indicates that the displacement is away from the satellite in the LOS direction. The time series of PS pixels A, C, D and F, which are at the edge of the potential sliding blocks, show less clear long period surface displacement. However, regardless of whether the long period surface displacement of the area is stable, the short period surface variation is shown clearly by the red triangles of the selected PS pixels. The short period surface variation appears to be related to the variation of the dry and wet seasons, where precipitation is mainly concentrated from May to October.

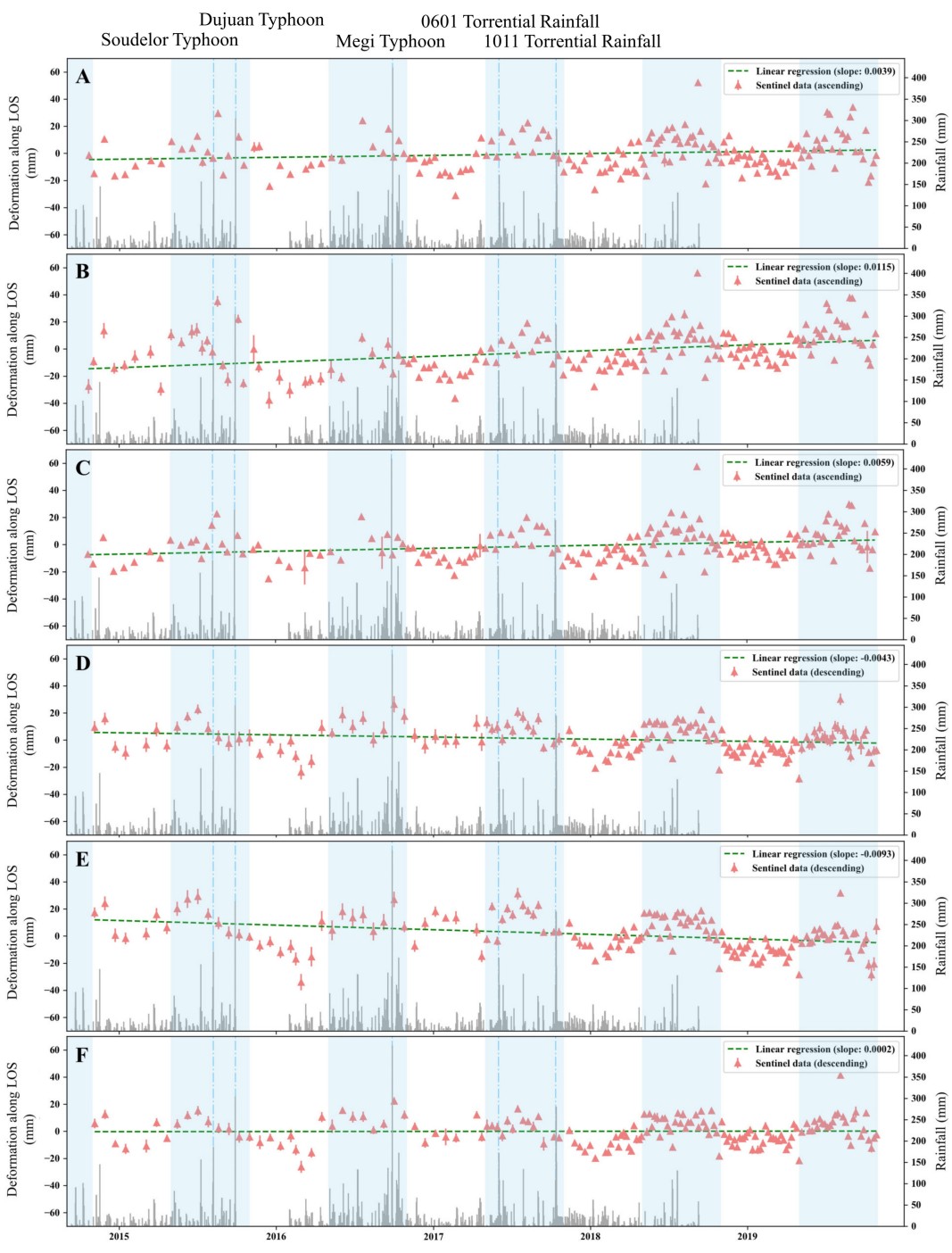

**Figure 4.** Time series are calculated from the average of the PS pixels within a 50 m radius of the selected PS pixel based on the ascending images and descending images from 2014-2019. The distribution of the selected PS pixels is shown in Figure 3. The time series of PS pixels A, B and C were derived from the results based on the ascending images. These time series show positive slopes, which indicate the displacements are toward the satellite along the LOS direction. The time series of PS pixels D, E and F were derived from the results based on the descending images. The negative slopes indicate the displacements are away from the satellite along the LOS direction. These time series show evident signals of long period surface displacement and clear short period surface variation. The total displacement amount was approximately 20–30 mm along the LOS direction. The orange triangles indicate the average of the selected PS pixels with an error bar of one standard deviation. The light blue background indicates the wet season periods, while the white background indicates the dry season periods. Precipitation data from 01 September 2014 to 11 September 2018 were derived from the rainfall station on the Huafan University campus, and a precipitation measurement gap existed from 10/2015 to 01/2016.

## 5. Discussion

### 5.1. Projected LOS Velocity and 2D Displacement Velocity Field

Figure 5 shows the results of the LOS displacement velocity projected onto the maximum slope direction. The coefficient C is strongly sensitive to small slope variations, while the real direction of movement is most likely more uniform. Thus, a smoother topography is required. Considering that the spatial resolution of Sentinel-1 for Taiwan is approximately 10 m, a DEM with a resolution of approximately 20 m was considered appropriate for the calculation of coefficient C. According to the average values of coefficient C derived from the ascending and descending orbital geometries of approximately 0.5 and 0.4, respectively, the LOS direction is able to detect approximately 40%–50% displacement along the maximum slope direction. Thus, the activity threshold of $V_{slope}$ is set as 2–2.5 times the LOS activity threshold of approximately $\pm 2$ mm/yr. The activity threshold of $V_{slope}$ is thereby set as $\pm 5$ mm/yr. Compared to the horizontal velocity field derived from the in situ monitoring points proposed by the previous study [29], the displacement direction and pattern of the velocity distribution are consistent with the result of $V_{slope}$ in this study. The maximum displacement velocity of $V_{slope}$ is approximately 25 mm/yr and mainly occurs at the center area of the large potential sliding block of the Huafan University campus. The maximum horizontal velocity of the in situ monitoring point is about 27 mm/yr [29]. Because the sensitivity percentage of detecting the real surface displacement by the SAR sensor could not be 100%, the value of $V_{slope}$ would be slightly underestimated. The displacement velocity, mostly lower than 10 mm/yr, was estimated at the western small sliding block. Moreover, the PS pixels provide more displacement information than in situ monitoring points in the lower area of the large sliding block.

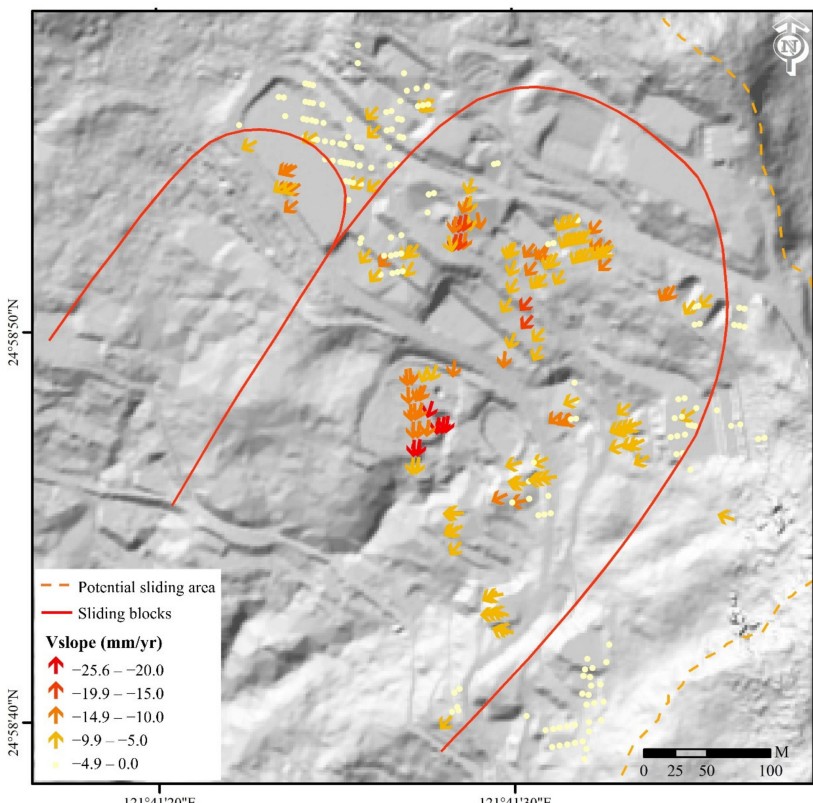

**Figure 5.** The projection of both ascending and descending LOS velocities along the downslope direction. The slow-moving phenomenon was significant, and the maximum displacement velocity along the downslope direction was approximately 25 mm/yr. The arrows indicate that the displacement was along the downslope direction. The background topography is shown by shaded relief imagery produced from a 2 m DEM.

The eastward and vertical displacement fields of the slow-moving landslide derived from the Sentinel-1 data are shown in Figure 6. According to the result of the eastward displacement field, the displacement of the two sliding blocks is consistently toward the westward direction. The maximum value of the westward displacement velocity is approximately 8 mm/yr. The vertical displacement velocity shows a pattern with maximum negative values of approximately 5 mm/yr, which means subsidence of the ground surface in the middle and upper parts of the two sliding blocks. The uplift of the ground surface is observed at the toe of the sliding blocks according to the vertical displacement velocity field. The uplift is interpreted as a compressional bulge, which is a common phenomenon of gravitational deformation [7]. A bulge of buckling material near the toe of the slope was created while the unstable body slid along a weak interlayer on the upper slope. This phenomenon is also consistent with the pattern of contractional strain from the horizontal strain analysis in the previous study of the Huafan University campus [29].

### 5.2. Seasonal Surface Fluctuation and Gravity-Induced Deformation

Generally, in the wet season or during heavy rainfall events, the increase in pore pressure causes a decrease in grain-to-grain friction and effective shear strength. Water mass loading can increase the gravitational driving force of an unstable slope. Thus, accelerated movement of a slow-moving landslide can be observed. The hydrology-driven seasonal acceleration of slow-moving landslides was observed by low temporal revisited SAR satellites, such as ALOS and ENVISAT. The seasonal acceleration has been described in previous studies [23–25].

In this study, the opposite slope value of the long period time series derived from the ascending and descending SAR images can be found due to the geometry of the satellite orbits. In the ascending case, the slope value of the long period time series is positive, which indicates that the long period surface displacement is toward the LOS direction. In contrast, the slope value of the long period time series derived from the descending SAR images is negative, which indicates that the surface displacement is away from the LOS direction. These results all indicate the surface displacement toward the south-southwest along the LOS direction. However, the same seasonal surface fluctuation in both the ascending and descending results can be detected in Figure 7 These time series show a strong correlation with precipitation. A seasonal interaction model for interpreting the combined effects of the pore water pressure and the water mass loading on the vertical surface displacement was proposed in our previous study [30]. According to the seasonal interaction model, the increased pore water pressure will cause expansion of the geological materials and the increased water mass loading will lead to compaction of the geological materials. The change of surface elevation reflects the overall amount of the expansion and compaction. Under this condition, the pore water pressure plays a predominant role in determining the vertical surface displacement, resulting in the surface displacement being in-phase with the precipitation of this study area. Thus, the slope surface will uplift in the wet season and subside in the dry season, which is different from the phenomenon of the hydrology-driven seasonal acceleration. In addition, due to the geometry of the satellite, the SAR sensor is more sensitive to the vertical displacement than to the SW-NE displacement. This study suggests that the main displacement component of the seasonal surface fluctuation in the study area is vertical. The uplift phenomenon in the wet season would be toward the satellite along the LOS direction, whether in the ascending or descending orbital geometry. Therefore, the seasonal surface fluctuation detected with the ascending and descending SAR images would be synchronized. Figure 7 clearly shows that the surface displacement of a slow-moving landslide involves long period gravity-induced deformation and short period seasonal surface fluctuation.

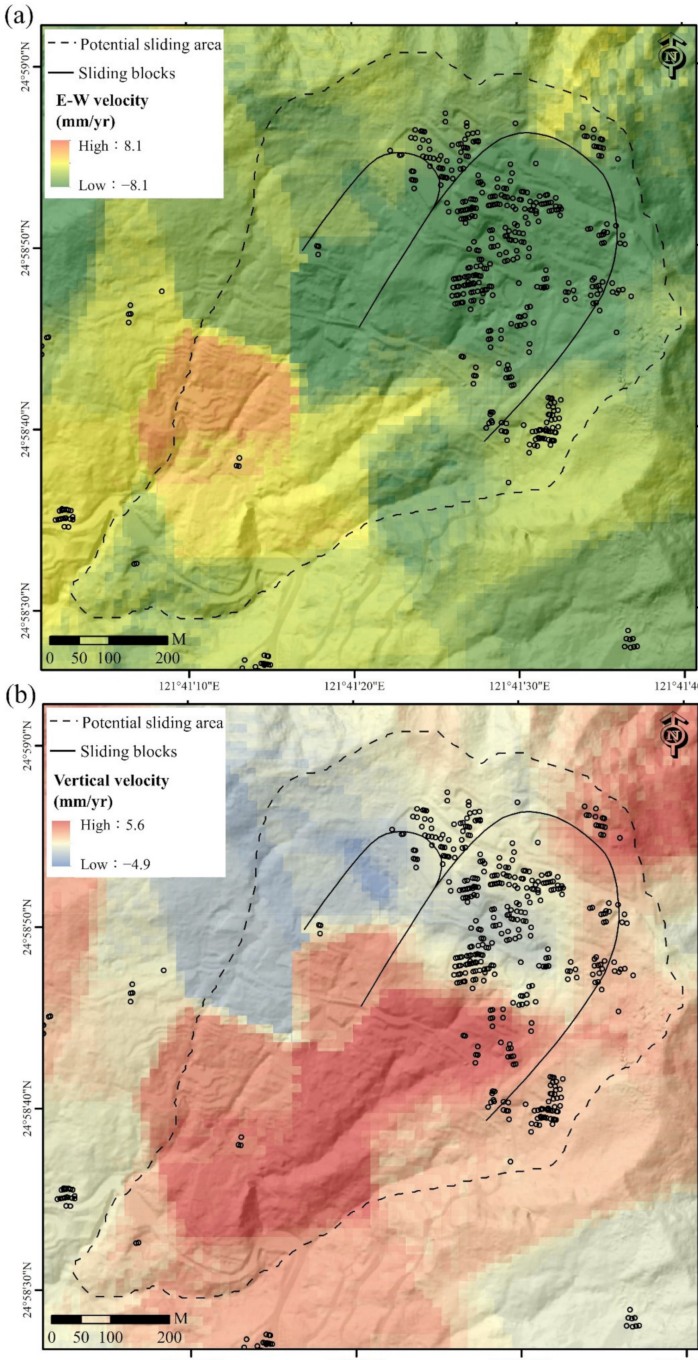

**Figure 6.** The E–W and vertical displacement fields were obtained from the PSInSAR results of two different satellite viewing geometries. (**a**) The surface displacement of the slope toward the west direction is clearly shown. Red indicates the surface displacement is toward the east, while green indicates the surface displacement is toward the west. (**b**) There is significant subsidence on the upper part of the slope and uplift on the toe of the sliding blocks. Red indicates subsidence of the surface, while blue indicates uplift of the surface. The background topography is shown by shaded relief imagery produced from a 2 m DEM.

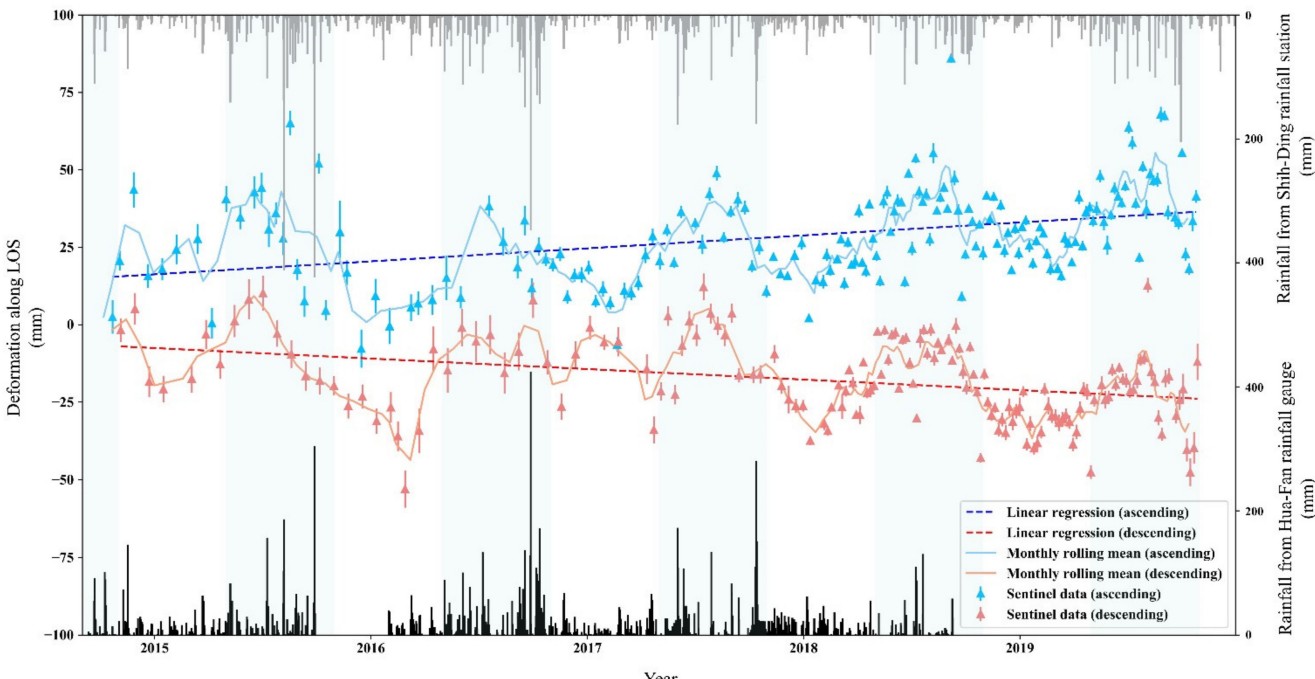

**Figure 7.** The comparison between time series derived from ascending and descending orbital geometries, seasonal surface fluctuation and gravity-induced deformation are clearly shown. The seasonal surface fluctuation is in-phase with the precipitation data. According to the two precipitation records derived from the Huafan University campus and the Shih-Ding rainfall stations with a distance of approximately 4 km, the trend of daily precipitation from the two rainfall stations is similar, but the amounts of daily precipitation are different. Thus, setting the precipitation threshold value of an early warning for possible landslide hazards should take care when using the data from the nearby Shih-Ding rainfall station.

There are two precipitation records derived from the Huafan University and the Shih Ding rainfall stations, denoted as black and gray, respectively. The Shih Ding rainfall station is approximately 4 km from the Huafan University campus. Although the trends of daily precipitation from these two rainfall stations are similar, the amounts of daily precipitation are different, which indicates that the distribution of the amounts of precipitation will change even in nearby areas. The precipitation threshold is very important for the early warning of an unstable slope. Unfortunately, the precipitation data at the Huafan University campus were recorded only until September 2018. The precipitation data derived from nearby rainfall stations can provide some reference information but should not be adopted directly for the precipitation thresholds during possible landslide movement emergency events.

### 5.3. Assessment of a Corner Reflector Installed at the Huafan University Campus

The whole potential sliding area is determined by the characteristics of the topography (area labeled by orange dashed line in Figure 1a), which is much larger than the Huafan University campus. However, PSInSAR cannot detect the surface displacement without persistent scatterers (such as buildings) from the Huafan University campus. Thus, a test of a trihedral corner reflector was set up in a garden, where there are no natural persistent scatterers. The intensity and coherence maps of SAR images, which were without (Figure 8a–c) or with (Figure 8d–f) the corner reflector, were analyzed to confirm that the corner reflector could be detected by the SAR sensor of the satellite. The brighter color in the intensity map indicates the greater reflected energy of the pixel, and the brighter color in the coherence map indicates the higher coherence value between two SAR images of that pixel. Due to the uniform pixel spacings in the azimuth and range of the SLC data and the projection of the data into a coordinate reference system using resampling in the terrain correction processing, the reflected signal of the corner reflector

presents a rectangular shape in the intensity map. The geocoding error may cause the position of the corner reflector to not be present at the middle of the brighter rectangle in the intensity map. According to the analysis, the intensity and coherence values were enhanced significantly and could be detected by the SAR sensor after the corner reflector was placed. The preliminary time series result of the corner reflector from 27 February 2020 to 4 June 2020 is shown in Figure 9. The surface displacement toward the LOS direction of the ascending orbital geometry is clearly observable. The displacement velocity of approximately 39 mm/yr is greater than the relatively stable long period moving velocity on the upper part of the Huafan University campus. This is due to the time series being derived from the very short monitoring period, which is within the seasonal fluctuation, using the corner reflector. Few results of applying corner reflectors for potential landslide monitoring have been published in the literature in Taiwan. Thus, this preliminary result provides the feasibility of applying corner reflectors in potential landslide areas to increase the stable monitoring points where the persistent scatterers are insufficient in Taiwan.

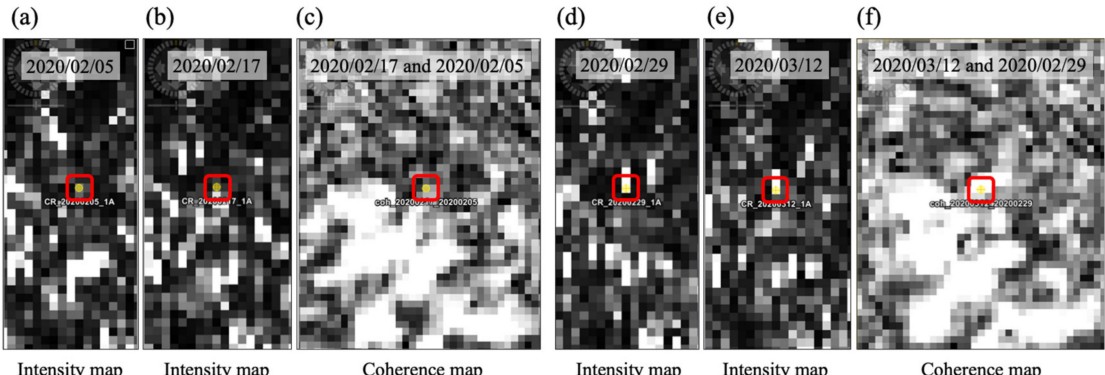

**Figure 8.** The intensity and coherence maps of the SAR images. The intensity and coherence values were enhanced significantly after the corner reflector was placed. The intensity maps of the SAR images acquired before the placement of corner reflector are (**a**) 05 February 2020 and (**b**) 17 February 2020. (**c**) The coherence map calculated between the SAR images derived on 05 February 2020 and 16 February 2020. The intensity maps of the SAR images acquired after the placement of corner reflector are (**d**) 29 February 2020 and (**e**) 12 March 2020. (**f**) The coherence map calculated between the SAR images derived on 29 February 2020 and 12 March 2020. The yellow circle indicates the position of the corner reflector.

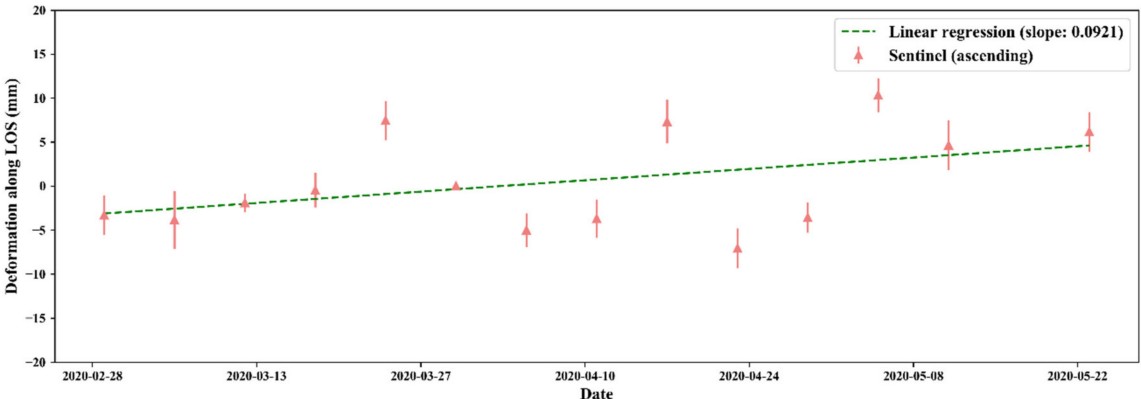

**Figure 9.** The time series showing the average values of the PS pixels in a 50 m radius of the corner reflector from 27 February 2020 to 4 June 2020 of the ascending orbital geometry. The surface displacement velocity detected by the corner reflector is approximately 39 mm/yr, which indicates short period seasonal fluctuations and is higher than the long period gravity-induced deformation. The orange triangles indicate the average of the selected PS pixels with an error bar of one standard deviation. The green dashed line indicates the linear regression of the time series.

## 6. Conclusions

The multitemporal interferometry techniques have been widely used to monitor surface deformation in several studies. The PSInSAR method, which is one of the multitemporal interferometry techniques, shows the ability to monitor slow-moving landslides effectively for a long period of time in this study. The major findings of this PSInSAR analysis based on the Sentinel-1A/B data are listed as follows:

1. The surface displacement pattern derived from the PSInSAR in this study is consistent with the active areas of the two sliding blocks, which were identified by field surveys and in situ monitoring data on the Huafan University campus in northern Taiwan. The PSInSAR method can compensate for the lack of in situ measurements of surface displacement.

2. According to the time series from the PS pixels, the movement of the slow-moving landslide can be divided into long period gravity-induced deformation and short period seasonal surface fluctuation. Based on the geological and hydrological conditions of this study area, the effect of pore water pressure predominated over the effect of water mass loading. The seasonal surface fluctuation is in-phase with precipitation.

3. By comparing the precipitation data from the campus and a nearby rainfall station, it was noted that the distribution of the amount of precipitation will change even in nearby areas. Therefore, the precipitation data derived from nearby rainfall stations should not be adopted directly as precipitation thresholds for an emergency evacuation due to possible landslide hazards. The installation of a rainfall gauge at the precise location of a potential landslide should be considered for evaluating possible landslide hazards.

4. The preliminary results of the corner reflector in this study provide the feasibility of applying corner reflectors in potential landslide areas in Taiwan where persistent scatterers are insufficient.

**Author Contributions:** Conceptualization: C.-Y.L. and Y.-C.C.; Formal analysis: C.-Y.L.; Visualization: C.-Y.L., Y.-C.C. and J.-C.H.; resources: C.-H.T., C.-H.L. and C.-H.C.; Writing—original draft: C.-Y.L.; Writing—review and editing: C.-Y.L., Y.-C.C. and J.-C.H.; supervision: Y.-C.C. and J.-C.H. All authors have read and agreed to the published version of the manuscript.

**Funding:** This research was supported by the Ministry of Science and Technology of Taiwan Nos. MOST109-2116-M-001-020 and MOST110-2116-M-001-014 and the thematic research project AS-TP-108-M08 of Academia Sinica.

**Data Availability Statement:** The SAR data used in this study can be accessed in a publicly archived datasets https://search.asf.alaska.edu/#/ at the Alaska Satellite Facility (ASF).

**Acknowledgments:** We thank the Central Geological Survey and Soil and Water Conservation Bureau in Taiwan for providing valuable information about the geological survey data in this study. The work is supported by the Ministry of Science and Technology of Taiwan and Academia Sinica. We also thank Ching-Jiang Jeng, General Affairs Chief of Huafan University, for supporting the slow-moving landslide investigations at the Huafan University campus.

**Conflicts of Interest:** The authors declare no conflict of interest. The funders had no role in the design of the study; in the collection, analyses, or interpretation of data; in the writing of the manuscript; or in the decision to publish the results.

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
