# Peer review of "Seasonal Surface Fluctuation of a Slow-Moving Landslide Detected by Multitemporal Interferometry (MTI) on the Huafan University Campus, Northern Taiwan"

_remotesensing, doi:10.3390/rs13194006_

Round 1

Reviewer 1 Report

This manuscript presents the seasonal surface fluctuation of a slow-moving landslide using the multitemporal interferometry at the Huafan University campus of Taiwan. The presentation of this manuscript seems quite interesting, however; it is only focusing on a regional area (at Huafan University campus) of the northern Taiwan. Therefore, the reviewer feels that it will be better to consider this paper as a “Case Study” rather than an original article. Moreover, following are a few overall commands, which may help to improve the quality of paper for considering in publication in future.

  1. Introduction

-Please highlight the major research problem and objectives of this study.

-In addition, please cite the relevant recent papers in this section.

  1. Background

-It will be better to change ‘Background’ into “Study Area”, and also add the location map of the study area for better understanding of the exact location of study to the reader.

- I think the content of the sub-section ‘2.2. Monitoring system and failure mechanism’ are related to the ‘Methodology’ and “Discussion’ section. Please think seriously about it.

  1. Methodology

- Please add the procedure of the overall work in a ‘flowchart’ or other illustration if it is possible. The current methodology section doesn’t well explain all steps of work.

  1. Surface displacement monitoring of the slow-moving landslide

-It will be better to restructure of this section as follows:

4.Results

4.1. Surface displacement monitoring of the slow-moving landslide

  1. Discussion

-Please compare your results and trends to other researchers’ relevant works.

-Justification and validation of the results of this study still seems missing.  

-Also, it will be better to add the failure mechanism of slow-moving landslide based on seasonal surface fluctuation.

  1. Conclusion

- Please list out the major finding of your study only.

Author Response

Our response is attached as a pdf file. 

Reviewer 2 Report

The paper is well written and the concepts and results are very well presented. 

A few suggestions:

Consider putting the reference to Figure 1 in the paragraph at the top of page 3

The figures are very useful. Unfortunately, in most of the them (e.g., Figure 1, 2, 3, 4, 6, 7, 9) the text is too small to read. Improving the legibility of the figures will greatly improve the quality of this paper.

In the description of Fig. 1, "DEMs" is used but not previously defined. I see that it is defined later in the text. It should probably be defined here.

Line 188: SLC is not defined

Line 222: incidence angles at of C-band satellites.

The findings of this research appear valid and it is clear that there are other sensors used to provide control for the results:

"To understand the failure mechanism and provide risk management, monitoring systems including 32 inclinometer casings, 32 standpipes for groundwater table monitoring 117 and 2 rainfall gauges have been installed since 2000. There are 15 tiltmeters installed on 118 the building walls, 48 strainmeters installed on the reinforcing bars, and 36 strainmeters 119 installed on the concrete of the buildings for monitoring the tilt situation of the structures. 120 In addition, a network of 295 ground monitoring points was established in 2001 for measuring the sliding behavior of the dip slope using conventional traverse surveying twice a 122 year"

Wherever possible, it would be beneficial to link your results to the results from this sensor network to illustrate their validity.

Conclusion: The title and introduction of the paper indicates that multitemporal interferometry is a key part of this study. The conclusion does not specifically mention the importance of this approach. It would be useful to include a sentence or two about the value of MTI in this research.

Author Response

Our response is attached as a pdf file.

Reviewer 3 Report

(1)The principle of multitemporal interferometry is less elaborated, and the formula related to the principle can be appropriately added.

(2)The projected LOS velocity and 2D displacement velocity field discussed can be put into the part of Methodology.

(3)The correlation between the precipitation data and the time series of surface deformation can be further quantified to determine the reliability of the conclusion.

(4)The parameters can be adjusted appropriately to improve the density of PS points and increase the reliability of the results.

(5)The precise leveling data or surface deformation data from others can be added to ensure the reliability of the results.

(6)The data of underground water level measurement can be added to explain the impact of seasonal precipitation on surface deformation.

(7)Among the two groups of landslides, the surface deformation information of the larger landslides is rich, and the PS points density of the smaller landslide is small, which has a certain impact on the conclusion of the experiment.

(8)The time density for the average of the selected PS pixels in Figure 9 is too small, and the law and stage of short-term seasonal fluctuation are not very obvious.

(9)It is better to add the surface deformation rate map around the study area, and combined with the topographic map and precipitation data, the displacement of the slow-moving landslide can be analyzed from a more overall perspective.

(10)The causes and trends of landslide sliding can be analyzed in more detail in combination with soil permeability.

Author Response

Our response is attached as a pdf file.

Round 2

Reviewer 1 Report

The authors have been addressed all my commands.

Reviewer 3 Report

The original manuscript has been sufficiently improved in the revised version. The contents are relatively complete, and the comments are addressed or well explained.